# An in Silico Approach to Identifying TF Binding Sites: Analysis of the Regulatory Regions of BUSCO Genes from Fungal Species in the *Ceratocystidaceae* Family

**DOI:** 10.3390/genes14040848

**Published:** 2023-03-31

**Authors:** Nomaswazi N. Maseko, Emma T. Steenkamp, Brenda D. Wingfield, P. Markus Wilken

**Affiliations:** Department of Biochemistry, Genetics and Microbiology, Forestry and Agricultural Biotechnology Institute (FABI), University of Pretoria, Pretoria 0083, South Africaemma.steenkamp@fabi.up.ac.za (E.T.S.); markus.wilken@fabi.up.ac.za (P.M.W.)

**Keywords:** promoters, motif discovery, *Ceratocystidaceae*

## Abstract

Transcriptional regulation controls gene expression through regulatory promoter regions that contain conserved sequence motifs. These motifs, also known as regulatory elements, are critically important to expression, which is driving research efforts to identify and characterize them. Yeasts have been the focus of such studies in fungi, including in several in silico approaches. This study aimed to determine whether in silico approaches could be used to identify motifs in the *Ceratocystidaceae* family, and if present, to evaluate whether these correspond to known transcription factors. This study targeted the 1000 base-pair region upstream of the start codon of 20 single-copy genes from the BUSCO dataset for motif discovery. Using the MEME and Tomtom analysis tools, conserved motifs at the family level were identified. The results show that such in silico approaches could identify known regulatory motifs in the *Ceratocystidaceae* and other unrelated species. This study provides support to ongoing efforts to use in silico analyses for motif discovery.

## 1. Introduction

The initiation of transcription is an integral part of gene expression and occurs within the promoter, the region directly upstream of genes [1,2]. The functional units of the promoter are called regulatory elements, which are conserved sequence motifs that act as binding sites for regulatory proteins such as transcription factors (TFs) [2,3]. These regulatory elements have been the focus of numerous studies, and efforts to understand gene regulation at the transcriptional level have focused on identifying and characterizing these DNA binding motifs.

The identification of motifs (commonly referred to as ‘motif discovery’) can be performed either in vitro or in silico. Initially, in vitro motif discovery relied on techniques exploiting physical protein–DNA interactions such as the electrophoretic mobility shift assay (EMSA) [4], DNase I protection (footprinting) assay [5] and South Western blotting [6]. However, with the increasing availability of whole genome sequences, motif discovery has moved to the realm of bioinformatics. In silico methods for motif discovery are based on creating sequence alignment profiles, after which the occurrence of each nucleotide is quantified to identify and further characterize any motifs that may be present [7]. Many tools to simplify motif discovery have been developed, including platforms such as TRANSFAC, MEME, Motif-Sampler and CONSENSUS [8]. These tools search for statistically overrepresented motifs that occur more often than expected by chance alone [9].

Motif discovery algorithms can either use de novo (or unbiased) searches or the searches that are informed by known regulatory motifs [10]. De novo motif discovery algorithms take advantage of the fact that regulatory elements tend to occur in non-coding regions, which show more sequence divergence than coding regions [11]. Among these divergent regions, regulatory elements would show higher sequence conservation due to a slower divergence rate in the regulatory regions [11]. These evolutionary patterns prompted the use of motif discovery strategies such as the ‘single gene, multiple species’ approach where the same gene is compared among multiple species, and the ‘multiple genes, single species’ approach assuming gene regulation is conserved in a single organism [12]. For example, the ‘multiple genes, single species’ approach was used to detect over-represented motifs in non-coding sequences from co-regulated genes in the yeast genome [13].

De novo motif discovery in fungi has mostly been successful in the model yeast species [3,14,15], although some work has been performed in non-model species as well. De novo TF motif discovery were reported in *Fusarium graminearum*, *F. verticillioides*, *F. oxysporum* and *F. solani* [16], all non-model filamentous ascomycetes. However, for many filamentous fungi, little to nothing is known about regulatory regions for transcription. This knowledge gap includes information about the architecture and regulatory element presence, as well as how these compare to previously characterized fungal regulatory regions and elements.

The current study aimed to determine whether in silico approaches could be applied to discover putative TF motifs from the *Ceratocystidaceae,* for which very little is known about transcriptional regulatory regions. This fungal family was chosen for its economic importance, which is underpinned by the large number of plant pathogenic fungi that belong to this group [17,18]. Additionally, full genome sequences are available for many species, lending itself to an in silico analyses. A total of 25 *Ceratocystidaceae* genomes across the family were used. A 1000 bp region upstream of the start codon was targeted as the putative regulatory region for 20 candidate genes. This region was analysed for the presence of conserved sequence motifs that could represent putative regulatory regions.

## 2. Materials and Methods

### 2.1. Gene Selection and Annotation for Motif Discovery

A selection of genomes representing 25 species across seven genera of the *Ceratocystidaceae* was used to identify unique motifs that could be potential transcriptional regulators. All genome sequences were obtained from the NCBI database (www.ncbi.nlm.nih.gov, [19]) (Table 1).

Genes were selected for analysis based on the results of an analysis using the Benchmarking Universal Single-Copy Orthologs (BUSCO) (Felipe A Simão, version 2.0.1, California, USA) tool [39]. To do this, each genome was subjected to a BUSCO analysis using genome mode and the Ascomycota_odb9 dataset, with *F. graminearum* as a reference species. The output included a table that stipulates whether each BUSCO ortholog (represented by the BUSCO ID) are complete, duplicated, fragmented, or missing in the analysed genome. Only complete genes that were present across all 25 genomes were chosen for further analysis. These were sorted alphabetically by BUSCO ID, and the first 20 genes were selected (Table 2).

To identify regions that would putatively contain promoter regions, the chosen genes were annotated in each genome. To do this, the GFF files generated by BUSCO were used to annotate the genes using the “Annotate with GFF file” plugin in CLC Genomics Workbench version 11.0 (Qiagen, Aarhus, Denmark). Following the annotation of the 20 selected genes, the first 1000 base pair (bp) upstream of the ATG start codon of each gene in each species was selected as the probable range for regulatory region presence. These 1000 bp sequences were grouped by gene and exported as a single FASTA files. In total, 20 such FASTA files were generated, with each FASTA file containing 25 different 1000 bp sequences, one from each of the 25 species. All these regions were subjects to a BLASTn search against the nt/nr nucleotide database [19] to evaluate whether any of these regions formed part of a different gene region, which could influence motif discovery. BLASTn parameters were adjusted to only use the “ascomycete fungi” subset of the dataset, to report a maximum of 10 target sequences, and to not report matches with an e-value of more than 0.01.

### 2.2. Motif Discovery

The Multiple Em for Motif Elicitation (MEME) tool [40] from the online interface of MEME suite (Timothy L. Bailey, version 5.2.0, Reno, NV, USA) [41] was used for the de novo prediction of motifs. The exported FASTA files were individually used as input files in MEME, and the number of motifs to identify were limited to 30 motifs per gene for each dataset. This number was chosen based on initial trial runs that limited the motif finder to 30 motifs and minimized the overlaps between different motifs. The motif discovery mode was set to classic; the sequence alphabet was set to DNA, RNA, or protein and the site distribution was set to zero or one occurrence per sequence (zoops). An additional parameter of motif presence in at least 80% of the species was also set. Only motifs that met all these parameters were retained for subsequent analyses.

To determine whether any motifs were repeated within the 1000 bp target regions, or were present in the 1000 bp regions of different genes, a Tomtom analysis was used. This analysis was performed from the online MEME interface [41]. The Tomtom tool compares a set of input sequences to a defined target set, and scores matches based on the *p*-value and E-value [42]. Here, the *p*-value is the probability that a random motif of the same width as the target would have an optimal alignment with a match score as good or better than the target’s, while the E-value is the expected number of false positives in the identified matches [42]. A database was constructed by including all motifs retained from the original MEME analysis, and was used to determine whether any of the motifs were unique. To do this, the individual motifs were combined into a single text document, and this combined dataset was used as both the query and a user-specified database in an “all-vs.-all” comparison in TOMTOM. Motifs were only considered significantly similar when both the reported E-value and *p*-value was ≤0.01.

The identified motifs were also mapped onto the corresponding 1000 bp sequences using the MAST tool [43] available in the web-based version of the MEME suite [41]. The MAST tool searches for given motifs in a set of sequences and provides a graphical map of the position of each motif. The maps were then analysed to evaluate the positional conservation of the motifs. The positional conservation of each motif was evaluated using two criteria: spatial position and order pattern. The spatial position indicates the distance between the motifs and the ATG start codon, while the order pattern referred to the order in which the motifs occurred in the 1000 bp region.

### 2.3. Motif Comparison

The resulting motifs from the MEME analysis were compared to known and characterised TF binding sites using the Tomtom tool set to use the JASPAR CORE (2018) and the JASPAR CORE fungi (2018) databases under the JASPAR non-redundant DNA category [44]. For each motif, the cut-off parameters were set at both an E-value and *p*-value of ≤0.01. Again, the *p*-value is the probability that a random motif of the same width as the target would have an optimal alignment with a match score as good or better than that of the target, while the E-value is the expected number of false positives in the identified matches [42].

### 2.4. Novel Motif Identification in Non-Ceratocystidaceae Species

To evaluate whether any motifs identified in the *Ceratocystidaceae* dataset might also be present in other fungi, four *Sordariomycetes* genomes outside of the *Ceratocystidaceae* were obtained from the NCBI Genbank database (www.ncbi.nlm.nih.gov/genbank, [45]) for analysis. These were the genome sequences of *Neurospora crassa*, *Sordaria macrospora*, *Ophiostoma novo-ulmi* and *Fusarium circinatum* (Table 1). To identify the homologs of the 20 genes used in this study from these four genomes, the relevant gene models from *Ambrosiella beaveri* were translated and used in a tBLASTn search against the target genomes. Regions producing significant BLAST hits (E ≤ 0.01) were used for de novo annotation on the online interface of the Augustus gene prediction software [46] using *F. graminearum* gene models as reference. Predicted genes were annotated using the “annotate with GFF file” plugin of the CLC Genomics Workbench. The predicted genes were then translated and subjected to a BLASTp search against the NCBI database to confirm the identity of each gene.

Following annotation, a 1000 bp sequence upstream of the ATG start codon of each of the 20 selected genes was isolated for each of the four genomes. These were then grouped according to gene and exported as single FASTA files. Each dataset was used as an input file in MAST and searched against the relevant set of motifs discovered in the *Ceratocystidaceae* genomes.

## 3. Results

### 3.1. Gene Selection for Motif Discovery

The BUSCO analysis of the 25 *Ceratocystidaceae* genomes identified between 574 and 1287 of the 1315 orthologs present in the Ascomycota_odb9 database as complete. *Ceratocystis albifundus* had the most complete orthologs (1287), while only 547 complete orthologs were identified from *Davidsoniella neocalidoniae*. A total of 454 orthologs were shared across all 25 genomes.

The 20 genes that were selected are involved in many different biological processes, including protein turnover (EOG092D01ZK), cell division (EOG092D03RY, EOG092D0454, EOG092D01QP), transcription and mRNA processing (EOG092D0AI2, EOG092D05X9, EOG092D05RI, EOG092D0564, EOG092D02YC, EOG092D00LL, EOG092D0124), translation and rRNA processing (EOG092D01J4, EOG092D0ACX), intracellular vesicle processing and trafficking (EOG092D03RC, EOG092D01YA, EOG092D01WX), DNA repair (EOG092D042R), chromatin remodelling (EOG092D01IY), cell signalling (EOG092D01MX) and cytoskeletal transport (EOG092D0072) (Appendix A).

The BLASTn analysis of the 1000 bp upstream region of all the genes of all species mostly did not show similarity to any other known sequence in the database (did not produce a BLAST hit, or the BLAST hit had an e-value of more than 0.01). In some cases, a BLAST hit was produced, but this was mostly to a chromosome region that was not associated with any coding region. In a minority of the regions, some blast hits were produced that matched a gene sequence. However, this was always for a single gene in a maximum of two species (of the 20 analysed). As the 1000 bp target regions were compared between all 20 species per gene, such a small overlap (in only 2 of the 20 species) would not significantly influence the identification of conserved motifs. The full dataset was used for subsequent analysis.

### 3.2. Motif Discovery in the Ceratocystidaceae Species

The MEME analysis identified motifs within the provided cut-off parameters for most, but not all the 1000 bp target regions. No motifs conforming to the target parameters were identified in the 1000 bp upstream regions for genes EOG092D01QP, EOG092D042R and EOG092D05RI. A total of 61 motifs were discovered across the 1000 bp target region for the remaining 17 genes. Although the minimum representation parameter for a motif was set as present in least 80% of the genomes, this value was higher in most cases, with 42 of the 61 discovered motifs present in at least 23 of the 25 genomes. Furthermore, despite the e-value and *p*-value cut-offs being set to <0.01, the resulting motifs had e-values and *p*-values well under this parameter, with the highest recorded e-value being 3.6 × 10^−12^ and the highest *p*-value being 8.47 × 10^−6^ (Appendix A). The number of motifs identified per gene varied from one motif per gene to ten motifs per gene, with a median of three motifs. These motifs ranged in length from 15 bp to 50 bp, with an average length of 43 bp and median of 49 bp (Appendix A).

The all-vs.-all Tomtom analysis produced 4 motif groups that contained between two and 13 motifs per group (Table 3). In total, 22 of the motifs were split into these groups, with 1 motif (EOG092D01YA_motif 8) present in two groups. A single gene could have motifs classified under different groups, such as motifs for gene EOG092D0124 present in groups 3 and 4, and motifs for gene EOG092D00LL present in groups 1, 2 and 3. Furthermore, for one gene (EOG092D00LL) multiple motifs were grouped into a single group and might be indicative of repeated copies of that motif in the target region. Ten of the 13 motifs from group 4 were enriched in adenine bases (Figure 1). These motifs were present in 6 of the 17 genes from group 4, with the genes EOG092D01YA, EOG092D01IY, EOG092D0072 and EOG092D03RC each having two adenine-rich motifs (Table 3). The remaining 39 motifs that were not matched to any groups were considered unique.

Mapping the motifs onto the relevant 1000 bp upstream region identified three recognizable distribution patterns (Figure 2). In pattern 1, the motif distribution showed a similar spatial position (the position from the ATG start codon) and the pattern (the order in which the motifs occurred) in all analysed regions, while pattern 2 was defined by good conservation in terms of the pattern but lacked spatial positional conservation. Lastly, pattern 3 was characterised by a lack of spatial and positional conservation. The distribution of conserved elements mostly followed pattern 3 (9 of the 17 genes), although two genes (EOG092D02YC and EOG092D00LL) matched pattern 1, and four genes (EOG092D0072, EOG092D01ZK, EOG092D01WX, EOG092D01J4) matched pattern 2. Genes EOG092D0454 and EOG092D01MX only had a single motif each, precluding the identification of any complex distribution patterns for these genes.
Figure 1The nucleotide sequence logos of the A-rich motifs, with Adenine in red, Guanine in yellow, Thymine in green and Cytosine in blue. The genes are represented by their BUSCO IDs. Individual motifs are also shown in higher resolution in Appendix A.
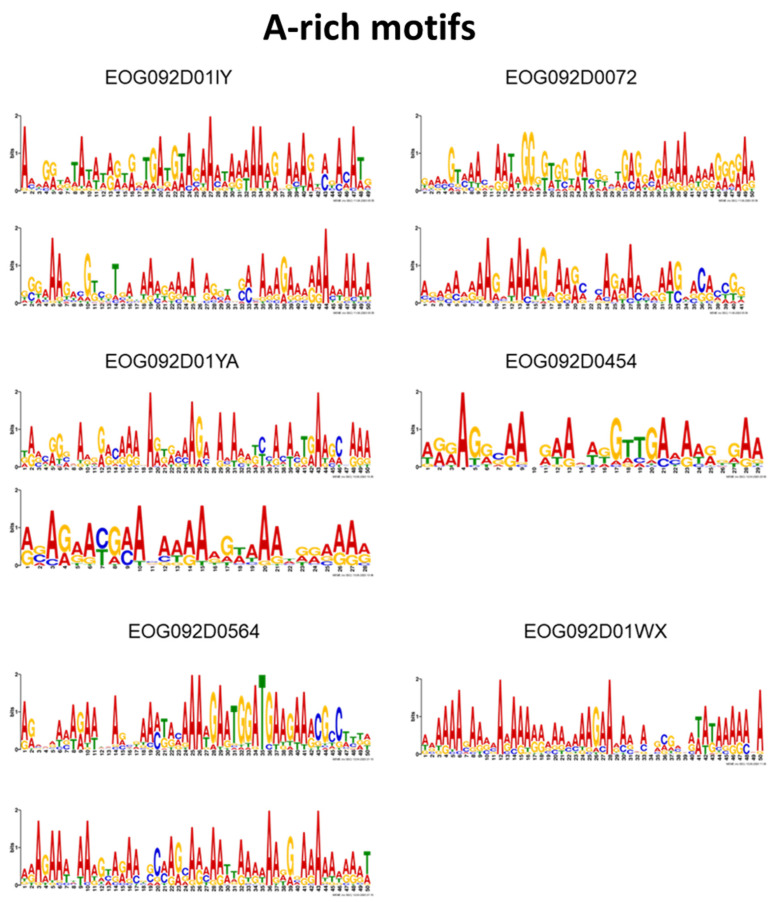



### 3.3. Motif Comparison to Known TF Binding Sites

When the 61 motifs discovered from the *Ceratocystidaceae* genomes were compared to known transcription factor binding sites from both the JASPAR core 2018 database and the JASPAR fungi 2018 sub-collection, 43 of the motifs showed matches while only 18 motifs did not have any matches. Among the matches, 25 returned hits to both databases, eight motifs returned hits to the JASPAR core 2018 database only and 10 motifs returned hits to the JASPAR fungi 2018 sub-collection only. The motifs were matched to multiple known TF binding sites, with most matches to the JASPAR core 2018 database (1 to 25 hits per motif) and less to the JASPAR fungi 2018 sub-collection (1 to 6 hits per motif) (Appendix A). All of the TF binding sites that matched in the JASPAR fungi 2018 sub-collection had *Saccharomyces cerevisiae* as the species of origin for the motif, while this varied in the JASPAR core 2018 dataset between *S. cerevisiae*, *Homo sapiens* and *Arabidopsis thaliana* which dominated the matches.

The identified *Ceratocystidaceae* motifs matched to 97 unique known TF binding sites belonging to 20 different TF classes in the JASPAR core 2018 database. The most represented class was the C2H2 zinc finger factors, comprising 66.2% of the alignment-matched motifs (Table 4). This is seven times more than the next most prevalent protein class, the MADS box factors with 9.6% of the known motifs aligning to the discovered motifs. The remaining 18 protein classes make up 24.1% of the aligned known motifs (Table 4).

A total of 61 unique TF binding sites matched from the JASPAR fungi 2018 sub-collection, belonging to 15 different TF classes. The C2H2 zinc finger factors were again the most represented class with 33.3% of the known motifs that aligned to the discovered motifs falling under this group (Table 5). The remaining 14 classes were more evenly distributed than those from the JASPAR core 2018 set, with the second most prevalent classes being fork head/winged helix factors and heat shock factors at 8.3% each, followed by other C4 zinc finger-type factors and tryptophan cluster factors at 6.9% each, and basic helix–loop–helix factors (bHLH), AT hook factors and C6 zinc factors all at 5.6%.

### 3.4. Identifying Novel Motifs in Non-Ceratocystidaceae

The 17 genes from which motifs were identified in the *Ceratocystidaceae* genomes were also successfully annotated in the genomes of *O. novo-ulmi*, *F. circinatum*, *S. macrospora* and *N. crassa*. Of the 17 genes investigated, MAST searches identified motifs corresponding to those discovered from the *Ceratocystidaceae* genomes in the 1000 bp upstream region of only 10 genes (Table 6). These included 17 motifs in the *O. novo-ulmi* genome, 11 motifs in the *F. circinatum* genome, 8 motifs in the *S. macrospora* genome and 6 motifs in the *N. crassa* genome (Figure 3). Despite *F. graminearum* being most closely related to Figure 4, most matches to the *Ceratocystidaceae* motifs were identified in *O. novo-ulmi*. The low number of hits per genes precluded the investigation for similarities in the distribution pattern of the motifs.

Only five (EOG092D0454_Motif 22, EOG092D0072_Motif 1, EOG092D0072_Motif 4, EOG092D01YA_Motif 8 and EOG092D01YA_Motif 18) of the 10 A-rich motifs identified from the *Ceratocystidaceae* genomes were found in the four non-*Ceratocystidaceae* genomes. These A-rich motifs were identified in the 1000 bp region of only three genes from the non-*Ceratocystidaceae* genomes as opposed to 10 genes from the *Ceratocystidaceae*.

## 4. Discussion

In this study, an in silico approach was used to identify potential promoter elements from the genomes of 25 *Ceratocystidaceae* species. This study is the first to attempt motif discovery using an in silico analysis at the genus level. The analysis identified 39 different conserved motifs present in the putative promoter region of 17 core BUSCO genes. Many of these motifs (70.5%) matched known TF binding sites associated with Eukaryotic promoter regions. The results of this study provide a framework for in silico motif discovery at a higher taxonomic rank, while also adding to the limited information currently available on regulatory motifs in the *Ascomycota* [3,16].

Although many motifs corresponding to known TF binding sequences were found, the C2H2 zinc finger factor motifs were the most abundant. The classical C2H2 domain is 28–30 aa in length and includes a β-hairpin stabilized by a zinc atom which allows C2H2 containing TFs to bind to long stretches of DNA [47]. The abundance of this group is not surprising considering that C2H2 zinc finger factors are one of the largest TF families [47,48]. In addition, an evaluation of TF prevalence in fungal genomes found that the C2H2 zinc finger factor superfamily was one of only 12 (out of 37) TF super families that were considered abundant in fungi [49].

A set of A-rich motifs were identified across many of the target regions, and these motifs shared sequence logo similarity when aligned in the Tomtom analysis. These sequence motifs also matched to the same known TF motifs, with 70% of these present in the non-*Ceratocystidaceae* genomes analysed in this study. A-rich sequences have long been associated with regulatory regions. This can be attributed to the ability of A-rich regions to easily bend, facilitating unwinding and allowing strand separation needed to initiate transcription [50].

It has been well documented that, for many genes, transcription initiation requires the sequential interaction of transcription factors (TFs) with their relevant transcription factor binding sites (TFBSs) [51]. The need for such a sequential interaction is thought to impose positional constrains on TFBSs, resulting in the positional conservation of these motifs [51,52]. Several motifs identified in this study did show positional conservation across the *Ceratocystidaceae* genomes, adding weight to the assumption that these are likely functional TFBs. For example, the pattern designated as pattern 1 in this study (similar spatial position and pattern in all analysed regions) is reminiscent of cis-regulatory modules (CRMs) which are clusters of motifs that co-occur on regulatory regions [53].

The in silico approach used in this study did not identify motifs in all 20 genes that were targeted, with no motifs identified in any representative species for three of the genes. One reason for this could be the methodology employed here. The approach relied on identifying overrepresented motifs in the aligned sequence set for each gene, making it possible that more divergent regulatory elements such as low-affinity binding sites and non-conserved functional sites could have been missed [54]. Another possibility is that the regulatory elements of these genes might be present outside of the 1000 bp regions used in this study [55,56]. Transcription can be regulated by gene-distal enhancers, and examples exist where enhancers can drive transcription independent of promoters. This has led to calls for a unified description of the promoters and enhancers [57,58,59]. This would make sense as enhancers and promoters share many characteristics such as similarities in their local structure (the presence of DNase hypersensitivity), they are often functionally interchangeable, rely on similar mechanisms for transcription initiation (both act as assembly points for RNA pol II and general transcription factors), both drive bidirectional transcription, and they display consistent histone modification patterns [58,60,61,62,63,64,65,66]. Disappointingly, only a small number of the discovered motifs were identified in the genomes of the four non-*Ceratocystidaceae* species, and might indicate that these elements are not true regulatory regions, but are regions of high-sequence conservation produced by common descent [67]. This can be dismissed for motifs that showed strong matches to known TF binding sites, and it might therefore be possible that other motifs might be novel conserved elements.

By making use of both the JASPAR (2018) and JASPAR fungi (2018) sub-collections for identifying known transcription factor binding sequences, some differences in the efficacy of these two databases were identified. The fungal sub-collection is largely biased towards *S. cerevisiae*, with 179 of the 184 TFs in the JASPAR (2018) fungi sub-collection originating from this species (Khan et al., 2018). Unsurprisingly, all of the matched TFs originated from this species, but the majority had no matches. This could indicate that the TF specific to these biding sites might not be present in *S. cerevisiae*, or that there is a high degree of sequence divergence due to the large evolutionary distance between *S. cerevisiae* and the *Ceratocystidaceae* species [68]. The JASPAR core (2018) database returned more hits per motif, and could be attributed to this subset having a higher level of cross-kingdom representation (Vertebrata, Nematoda, Insecta, Plantae, Fungi, and Urochordata) and more TF entries (1964 compared to 184 in the JASPAR fungi (2018) sub collection). Although *S. cerevisiae* was represented in both datasets, there were several discrepancies between the results from these two analyses, including variation in the identified TFs or matched in only one of the sub collections. Supplementing the JASPAR core 2018 database analysis with the JASPAR fungi (2018) sub collection produced the best results, but there is a clear need for updating and revising both datasets.

A de novo motif discovery approach such as that reported herein has a number of inherent risks that cannot easily be addressed. Identified similarities could be present due to chance alone, or due to sequence conservation unrelated to function. In this study, attempts were made to mitigate this risk by conducting a trial run of the MEME analysis to determine the optimal settings preventing overlaps in the identified motifs. In addition, only motifs that had wide representation among the *Ceratocystidaceae* genera were retained. Despite this, the functional analysis of the identified motifs would be needed to confirm their role (if any) in transcription [69,70,71]. Approaches for the genetic manipulation of *Ceratocystidaceae* species are actively being developed [72,73,74] and would be useful for future studies addressing these questions.

## Figures and Tables

**Figure 2 genes-14-00848-f002:**
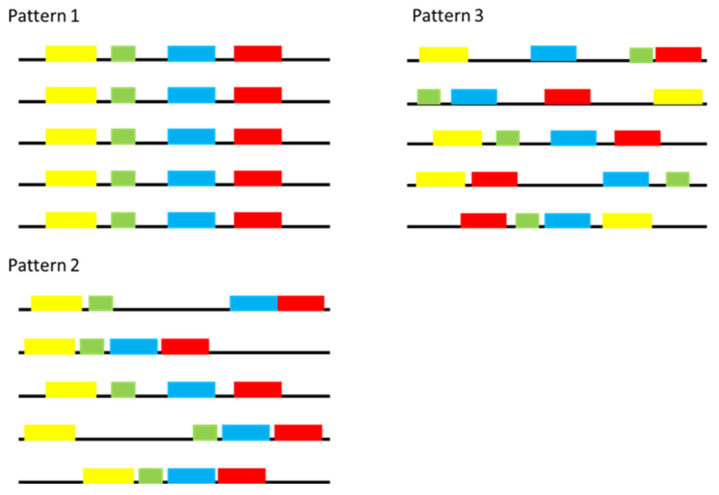
A diagrammatic illustration of the three different distribution patterns of the motifs across the different genes. Each black line represents a 1000 bp sequence upstream of the start codon for a gene, while coloured blocks represent different motifs present in this region. In Pattern 1, there is clear conservation in the spatial position and order of the motifs across the different genes. In Pattern 2, there was conservation in the order of the motifs only but conservation in the spatial position was lacking. There was no observed conservation in either motif order, or motif positioning in those motifs grouped in Pattern 3. This figure was modelled after the distribution patterns observed in this study, with full patterns provided in Appendix A.

**Figure 3 genes-14-00848-f003:**
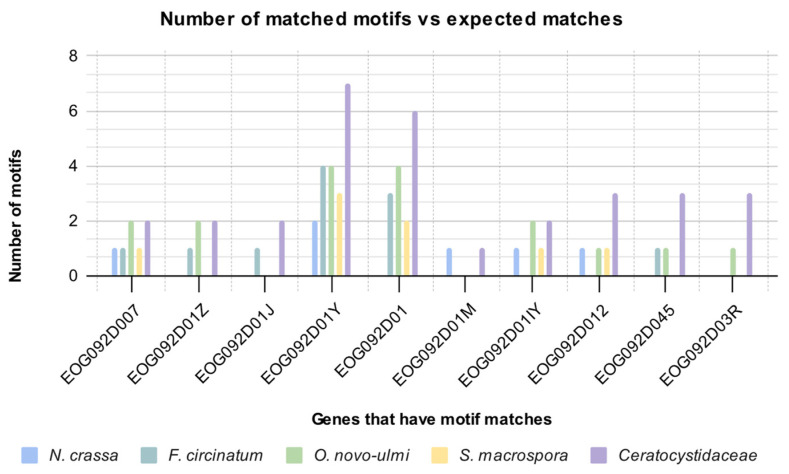
The distribution of motifs from the corresponding *Ceratocystidaceae* genes identified in non-*Ceratocystidaceae* species. The graph shows the number of motifs found in the 1000 bp region upstream of the ATG start codon in non-*Ceratocystidaceae* genomes that matched to corresponding motifs from *Ceratocystidaceae* genomes. The Ceratocystidaceae column represents the total number of motifs found in *Ceratocystidaceae* genomes per gene, and therefore represents the total number of possible matches.

**Figure 4 genes-14-00848-f004:**
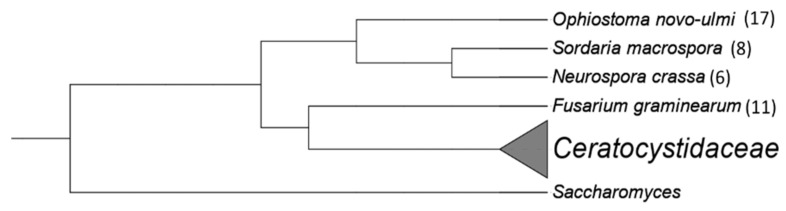
A schematic representation showing the phylogenetic relationship between the non-*Ceratocystidaceae* species and the *Ceratocystidaceae* family, rooted to *Saccharomyces*. The tree was constructed using the online interface phyloT based on the current taxonomic structure of the NCBI taxonomic database. Subsequent edits were made using iTol v6 (https://itol.embl.de/). The numbers in brackets indicate the number of motifs (from the *Ceratocystidaceae* genomes) that were found in each of the non-*Ceratocystidaceae* genomes.

**Table 1 genes-14-00848-t001:** Fungal genomes with references used in this study.

Species	Strain ^1^	Whole Genome Accession Number	References ^2^
*Ceratocystidaceae*			
*Ambrosiella beaveri*	CMW 26179 *	JARQWA010000000	Unpublished
*Ambrosiella cleistominuta*	CBS 141682 *	JABFIG010000000	[20]
*Ambrosiella xylebori*	CBS 110.61	PCDO01000000	[21]
*Berkeleyomyces basicola*	CMW 25440 *	PJAC01000000	[22]
*Ceratocystis adiposa*	CMW 2574 *	LXGU01000000	[23]
*Ceratocystis albifundus*	CMW 17620	JSSU01000000	[24]
*Ceratocystis fimbriata*	CBS 114723	APWK03000000	[25]
*Ceratocystis eucalypticola*	CMW 9998	LJOA01000000	[26]
*Ceratocystis harringtonii*	CMW 14789	MKGM01000000	[27]
*Ceratocystis manginecans*	CMW 17570	JJRZ01000000	[28]
*Ceratocystis smalleyi*	CMW 14800	NETT01000000	[29]
*Davidsoniella australis*	CMW 2333 *	RHLR01000000	[30]
*Davidsoniella eucalypti*	CMW 3254 *	RMBW01000000	[31]
*Davidsoniella neocaledoniae*	CMW 26392	RHDR01000000	[30]
*Davidsoniella virescens*	CMW 17339	LJZU01000000	[26]
*Endoconidiophora laricicola*	CBS 100207	LXGT01000000	[23]
*Endoconidiophora polonica*	CBS 100205	LXKZ01000000	[23]
*Huntiella bhutanensis*	CMW 8217	MJMS01000000	[27]
*Huntiella decipiens*	CMW 30855	NETU01000000	[32]
*Huntiella moniliformis*	CBS 118127	JMSH01000000	[28]
*Huntiella omanensis*	CMW 11056	JSUI01000000	[24]
*Huntiella savannae*	CMW 17300	LCZG01000000	[33]
*Thielaviopsis ethacetica*	JCM 6961	BCFY01000000	[34]
*Thielaviopsis musarum*	CMW 1546	LKBB01000000	[26]
*Thielaviopsis euricoi*	JCM 6020	BCHJ01000000	[34]
non-*Ceratocystidaceae*			
*Neurospora crassa*	OR74A	AABX03000000	[35]
*Sordaria macrospora*	k-hell	CABT02000000	[36]
*Fusarium circinatum*	FSP 34	AYJV02000000	[37]
*Ophiostoma novo-ulmi*	H327	AMZD01000000	[38]

^1^ Genome not obtained from NCBI indicated with asterisk. ^2^ Genomes not published indicated with hyphen.

**Table 2 genes-14-00848-t002:** BUSCO IDs and corresponding gene names.

BUSCO ID	Gene Name
EOG092D0072	Dynein heavy chain (*Penicillium* sp. 2HH)
EOG092D00LL	Pre-mRNA-processing-splicing factor 8 (*Penicillium* sp. 2HH)
EOG092D0124	General negative regulator of transcription subunit 1 (*Penicillium* sp. 2HH)
EOG092D01IY	SNF2-related protein (*Penicillium* sp. 2HH)
EOG092D01J4	U3 snoRNP protein (*Arthrobotrys oligospora*)
EOG092D01MX	Phospholipase D family protein (*Penicillium* sp. 2HH)
EOG092D01QP	Cell morphogenesis protein PAG1 (*Penicillium* sp. 2HH)
EOG092D01WX	phosphatidylinositol-4- kinase (*Arthrobotrys oligospora*)
EOG092D01YA	Clathrin, heavy chain (*Penicillium* sp. 2HH)
EOG092D01ZK	E3 ubiquitin-protein ligase listerin (*Penicillium* sp. 2HH)
EOG092D02YC	Transcriptional regulatory protein sin3 (*Arthrobotrys oligospora*)
EOG092D03RC	UDP-glucose:glycoprotein glucosyltransferase (*Penicillium* sp. 2HH)
EOG092D03RY	Sister chromatid cohesion protein 2 (*Arthrobotrys oligospora*)
EOG092D042R	DNA repair protein rad50 (*Arthrobotrys oligospora*)
EOG092D0454	Anaphase-promoting complex subunit 1 (*Arthrobotrys oligospora*)
EOG092D0564	THO complex subunit 2 (*Arthrobotrys oligospora*)
EOG092D05RI	PAN2-PAN3 deadenylation complex catalytic subunit PAN2 (*Penicillium* sp. 2HH)
EOG092D05X9	Transcription elongation factor spt6 (*Helotiales* sp. DMI_Dod_QoI)
EOG092D0ACX	Elongation factor EF-Tu (*Penicillium* sp. 2HH)
EOG092D0AI2	MIFG and Upf2 domain-containing protein (*Penicillium* sp. 2HH)

**Table 3 genes-14-00848-t003:** Results of the all-vs.-all comparison using Tomtom, which attempted to group the different motifs. Among the motifs, 4 groups were identified that shared motif identity. The motifs and genes in bold are present across multiple groups.

Group Number	Genes (BUSCO ID)	Matched Motifs ^1^
1	EOG092D01MX	Motif 28
	**EOG092D00LL**	Motif 1
	**EOG092D01YA**	**Motif 8** ^2^
2	**EOG092D00LL**	Motif 2, Motif 3, Motif 5, Motif 6
	EOG092D03RY	Motif 18
3	**EOG092D0124**	Motif 10
	**EOG092D00LL**	Motif 4
4	EOG092D0454	Motif 22 ^2^
	EOG092D0564	Motif 2 ^2^, Motif 3 ^2^
	**EOG092D01YA**	**Motif 8** ^2^, Motif 18 ^2^
	EOG092D01IY	Motif 9 ^2^, Motif 4 ^2^
	EOG092D01WX	Motif 7 ^2^
	EOG092D0072	Motif 1 ^2^, Motif 4 ^2^
	**EOG092D0124**	Motif 12
	EOG092D03RC	Motif 9, Motif 14

^1^ Motif numbers retained from the original MEME analysis for motifs that met the cut-off parameters. ^2^ Adenine rich motifs (see Figure 1).

**Table 4 genes-14-00848-t004:** The distribution of TF classes for motifs that returned matches to the JASPAR core 2018 database.

Known TF Classes	Number of Matched Motifs
C2H2 zinc finger factors	151
MADS box factors	22
Fork head/winged helix factors	11
Tryptophan cluster factors	8
Other	7
Helix–turn–helix	6
AP2/ERF domain	5
B3 domain	3
A.T hook factors	2
Heat shock factors	2
Basic helix–loop–helix factors (bHLH)	2
APSES-type DNA-binding domain	1
C6 zinc cluster factors	1
Other C4 zinc finger-type factors	1
Other α	1
Homeo domain factors	1
High-mobility group (HMG) domain factors	1
Basic leucine zipper (bZIP)	1
Copper-fist DNA-binding domain	1
Rel homology region (RHR) factors	1

**Table 5 genes-14-00848-t005:** The distribution of TF classes for motifs that matched to the JASPAR fungi 2018 sub-collection.

Known TF Classes	Number of Matched Motifs
C2H2 zinc finger factors	24
Heat shock factors	6
Fork head/winged helix factors	6
Tryptophan cluster factor	5
Other C4 zinc finger-type factors	5
Basic helix–loop–helix factors (bHLH)	4
A.T Hook factors	4
C6 zinc cluster factors	4
Copper-first DNA-binding domain	3
High-mobility group (HMG) domain factors	3
Heteromeric CCAAT-binding	2
Basic leucine zipper factors (bZIP)	2
Homeo domain factors	2
MADS box factors	1
APSES-type DNA-binding domain	1

**Table 6 genes-14-00848-t006:** Motifs discovered from the *Ceratocystidaceae* that were present in non-*Ceratocystidaceae* species.

Species	Motif
**EOG092D0072**	
** *N. crassa* **	Motif 1
** *F. circinatum* **	Motif 4
** *O. novo-ulmi* **	Motif 1, Motif 4
** *S. macrospora* **	Motif 1
**EOG092D01ZK**	
** *F. circinatum* **	Motif 3
** *O. novo-ulmi* **	Motif 2, Motif 3
**EOG092D01J4**	
** *F. circinatum* **	Motif 6
**EOG092D01YA**	
** *N. crassa* **	Motif 1, Motif 2
** *F. circinatum* **	Motif 2, Motif 5, Motif 9, Motif 18
** *O. novo-ulmi* **	Motif 2, Motif 5, Motif 8, Motif 18
** *S. macrospora* **	Motif 3, Motif 5, Motif 8
**EOG092D01WX**	
** *F. circinatum* **	Motif 1, Motif 2, Motif 4
** *O. novo-ulmi* **	Motif 1, Motif 2, Motif 4, Motif 7
** *S. macrospora* **	Motif 4, Motif 9
**EOG092D01MX**	
** *N. crassa* **	Motif 28
**EOG092D01IY**	
** *N. crassa* **	Motif 4
** *O. novo-ulmi* **	Motif 4, Motif 9
** *S. macrospora* **	Motif 4
**EOG092D0124**	
** *N. crassa* **	Motif 1
** *O. novo-ulmi* **	Motif 1
** *S. macrospora* **	Motif 1
**EOG092D0454**	
** *F. circinatum* **	Motif 22
** *O. novo-ulmi* **	Motif 22
**EOG092D03RY**	
** *O. novo-ulmi* **	Motif 5

## Data Availability

Publicly available datasets were analysed in this study. These data can be found at https://www.ncbi.nlm.nih.gov/ using the information provided in the manuscript.

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
