# Peer review of "An in Silico Approach to Identifying TF Binding Sites: Analysis of the Regulatory Regions of BUSCO Genes from Fungal Species in the Ceratocystidaceae Family"

_genes, 2023, doi:10.3390/genes14040848_

Round 1

Reviewer 1 Report

In the paper “An in-silico approach to identifying TF binding sites: analysis of the regulatory regions of BUSCO genes from fungal species in the Ceratocystidaceae family”, the authors analysed the 1kb upstream region for 20 single copy BUSCO genes in 25 species of the Ceratocystidaceae family of fungi and 4 outgroup species of fungi, specifically looking for enriched motifs with MEME and Tomtom.

Overall, I think it’s a good idea and a valuable contribution to the field. However, I would like further detail on the p-value and e-value cutoffs used - as these seem very permissive to false positives. It would be good to know if the results are still found with more strict cut-offs? And perhaps the individual motif p-values can also be reported. Furthermore, I would like to see some quality control addressing possible other genomic features, including additional coding regions, that could be captured by the approach taken. I think a simple approach would be for each of the motifs to simply be BLASTn to the nr database, and confirm that they are not subsets of any other genes.

Minor comments:

Line 85: “This table was manipulated in Excel® to retain only complete genes that were present across all 25 genomes.” – it’s unnecessary to mention Excel here. I’d replace this by simply stating that only complete genes across all 25 genomes were chosen for further work”

Methods: MEME and Tomtom – please ensure that all parameters, including default parameters are listed.

Line 114: “A database was constructed by including all motifs retained from the original MEME analysis. An “all-vs-all” _comparison…” – you’ve not specifically mentioned how this database was made or the comparisons made, I assumed on first read this was using BLASTn or p? Can you give further clarification on this point. I think this might be an option in Tomtom?

Line 116: “The default E-value (E-value>10)”. Default according to which program? Also, do you really mean greater than 10?! Or 1e-10? Greater than 10 sounds very permissive, as does p-value < 0.1? Can you comment on why you used these values, that do not seem very strict against false positives.

Table 3 is still a little confusing – and could have further clarification in the legend, What does the group number signify? And the motif numbers – what do they relate to? Are those given somewhere else to look up? Also could you include any information about the patterns you describe in figure 2 for each of these genes?

Figure 3 could do with some work – I suggest that this information would be clearer as a table rather than these pie charts – especially given the large number of categories, and subsequent difficulty with matching colours to descriptions.

Figure 4 – this is not a very aesthetic graph. Could it be remade in R?! Also, there should be axis titles – especially the y-axis! I don’t understand what the “Expected” category is or what it is based on?

 Figure 5: The species of Saccharomyces needs to be included – and why is there no indication of motifs found in this species? Also, you should include a scale bar and a description of what is meant by branch lengths in the legend. Also, are you sure this is a cladogram? How does phyloT construct trees? You should include more information about this.

Line 315: “The in-silico approach used in this study did not identify motifs in all 20 genes used targeted in this study.” – there must be a typo in this sentence – it doesn’t fully make sense to me. How many genes, or which genes had no TFBS identified?

Reviewer 2 Report

With the development of fungal genome sequencing technology, more and more genomic data are published in public databases. It is an important task to analyze the data information in order to understand the gene expression pattern. Transcription factors play an important role in the regulation of gene expression. The accurate analysis of transcription factor binding sites is an important part of functional genome research, but it usually takes a lot of time and money. This paper provides a set of in-silico approach to analyze these regions, which provides support for further study of gene expression.

Author Response

No requests for revisions were made